# Research on Measurement of Tooth Profile Parameters of Synchronous Belt Based on Point Cloud Data

**DOI:** 10.3390/s22176372

**Published:** 2022-08-24

**Authors:** Zijian Zhang, Mao Pang, Chuanchao Teng

**Affiliations:** School of Mechanical and Energy Engineering, Zhejiang University of Science and Technology, Hangzhou 310012, China

**Keywords:** synchronous belt, 3D measurement, normal vector, point cloud segmentation

## Abstract

Accurately detecting the tooth profile parameters of the synchronous belt is crucial for the transmission’s load distribution and service life. However, the existing detection methods have low efficiency, are greatly affected by the manual experience, and cannot realize automatic detection. A measurement method based on point cloud data is proposed to solve this issue. The surface space points of the synchronous belt are acquired by a line-structured light sensor, and the raw point clouds are preprocessed to remove outliers and reduce the number of points. Then, the point clouds are divided into plane and arc regions, and different methods are used for fitting. Finally, the parameters of each tooth are calculated. The experimental results show that the method has high measurement accuracy and reliable stability and can replace the original detection method to realize automatic detection.

## 1. Introduction

### 1.1. Measurement of Synchronous Belt

The synchronous belt is an important part of power transmission [1]. It has the advantages of an accurate transmission ratio, compact structure, and good wear resistance [2]. It is widely used in machine tools, light industries, chemical industries, automobiles, and other industries [3,4]. The geometric structure of the synchronous belt plays an important role in the load transmission, and the deviation of the tooth profile parameters will affect the service life and reliability. A proper tooth profile will reduce the interference between the synchronous belt and the sprocket, causing a better distribution of the load [5,6]. Conversely, the uneven load will increase the wear of the tooth surface and generate noise and vibration during operation, which leads to mechanical failures more frequently [7]. Therefore, it is very important to detect the tooth profile parameters of the synchronous belt.

The tooth profile deviation of the synchronous belt has several factors [8]: the wear of the mold, the dirt of the mold, the molding temperature, etc. During the production process, each tooth of the synchronous belt needs to be inspected. The current detection methods are mainly [9] the projection method and the manual method. Both methods require human operation, which is greatly affected by the manual experience and cannot work for a long time. The detection efficiency is low, and the cost is high.

### 1.2. Three-Dimensional Measurement

The definition of 3D measurement: acquiring the 3D data of the object and measuring the object in all directions. The current methods of 3D measurement can be classified as contact and non-contact [10]. Contact measurement methods include manual measurement using fixtures and three-coordinate measuring machines [11]. However, these methods have low measurement efficiency and limited measurement range, which means it is difficult to measure large-sized and special-shaped parts. Meanwhile, these methods need to contact the target to be measured, which easily leads to scratches and deforms the surface of non-rigid parts.

With the continuous development of computer technology and optical equipment, various non-contact 3D measurement methods continue to emerge, such as projected structured light [12], laser rangefinders [13], visual inspection technology [14], etc. Among these non-contact measurement methods, the structured light projection has become a prevalent non-contact 3D shape measurement technology [15]. Essentially, the 3D measurement method of structured light detects the visible surface of the object by configuring an additional light source generator to form the 3D shape of the object. The output of a 3D imaging system is usually a set of points with (*x*, *y*, *z*) coordinates for each measurement point in a Cartesian coordinate system. The advantages of the method are the simple structure, high measurement accuracy, and better stability. It is widely used in industrial inspection, agricultural inspection, 3D profile measurement, and other fields [16,17,18,19,20].

To rapidly obtain information on the 3D shape of the gear tooth flank, a 3D point cloud measurement system based on a line structure light sensor was proposed by Guo [21]. The measured 3D point cloud data were used to calculate the profile error and pitch error. The results showed that the 3D point cloud measurement system could perform rapidly and accurately in the 3D measurements of gears. Long [22] aimed to ensure the quality of aircraft assembly, and a method based on an unstructured scanned point cloud was proposed. Their method first detected and segmented the seam area based on the local point density discrepancy. A projection operation is applied based on the segmentation results to convert 3D measurement to 2D measurement to reduce the computational complexity and improve the accuracy of the measurement. Fan [23] proposed an approach to extracting curve profiles from a scanned point cloud. A slice set was constructed to extract section line points from the point cloud. Based on the normal angle of adjacent points, three methods were proposed to identify the curve-profile points on the three kinds of section lines. The results showed that extracted curve-profile points deviate from reference data within 0.1 mm with standard deviations smaller than 0.07 mm. Xu [24] proposed a method of identifying the longitudinal rip through 3D point cloud processing. The results indicated that this method could identify the longitudinal rip accurately in real-time and simultaneously characterize it.

### 1.3. Motivation

As can be seen, the 3D point cloud measurement methods have been widely used, and there have been better developments in size measurement. This work aims at the problems of low detection efficiency and human factors in the detection of the tooth profile parameters of synchronous belts. A measurement method based on point cloud data is proposed. Specifically, a line-structured light sensor is used to acquire point clouds on the surface of the synchronous belt. Remove outliers and simplify the point cloud through pre-preprocessing. Then, the point clouds are divided into plane and arc regions. Fitting models suitable for them are established by different methods. Finally, the detection of each tooth of the synchronous belt is completed.

Overall, our contributions are summarized as follows:We propose an effective method to detect the tooth profile parameters of synchronous belts from 3D point clouds and establish a point cloud acquisition system on the surface of synchronous belts;We design a point cloud segmentation algorithm based on the angle of the normal vector, which accurately obtains the measurement area of the tooth surface arc and establishes a fitting model;The 3D point cloud measurement system has better stability, and the measurement error is less than 0.03 mm.

This paper is organized as follows: Section 2 describes the proposed method in detail. The results of these tests are presented in Section 3 and they are discussed in Section 4. Finally, the paper is concluded in Section 5.

## 2. Methods

### 2.1. Methodology of Synchronous Belt Geometry Measurement

The detected object is an arc tooth of the synchronous belt, the tooth shape code is High Torque Drive (HTD). The pitch (Pb) of the synchronous belt will cause a pitch error during the meshing process of the synchronous belt and the synchronous wheel. The belt tooth cannot fully enter the gear tooth slot if the pitch error is too large, resulting in an unstable transmission system. In severe cases, the phenomenon of tooth climbing and tooth skipping will occur, which leads to the interruption of the power transmission. The tooth height (ht) and belt height (hs) of the synchronous belt also affect its strength and load-carrying capacity. It is important to detect the parameters of the synchronous belt.

Measurement plan: Pb is the distance between the center of the arc at the top of the tooth, so the Pb measurement needs to obtain the position of the centers; ht is the distance from the center to the plane B and plus the arc radius. The hs data can be obtained in the same way. The parameters of the synchronous belt are shown in Figure 1.

#### 2.1.1. Point Cloud Model

An object contour point cloud acquisition system based on the principle of laser triangulation [25] is shown in Figure 2. The laser is vertically irradiated on the surface of the measured object and diffusely reflected [26]. The reflected light is projected on the photosensitive chip by the camera lens. The imaged data signal will be transmitted to the controller and stored as a point cloud with coordinate information. With the movement of the mobile platform, the point clouds of the measured object are obtained.

As shown in Figure 2, $O_{w}\text{-}x_{w}y_{w}z_{w}$ is the measurement coordinate system, that is the world coordinate system; where the $O_{w}\text{-}x_{w}y_{w}$ plane is parallel to the working plane of the mobile platform, $O_{w}\text{-}x_{w}z_{w}$ is coplanar with the structured light plane and the $O_{w}y_{w}$ direction is consistent with the movement of the mobile platform. $O_{c}\text{-}x_{c}y_{c}z_{c}$ is the camera coordinate system. The $O_{c}$ is the imaging perspective center. $O_{p}\text{-}uv$ is the pixel coordinate system, which is the image plane of the camera. $O_{i}\text{-}xy$ is the image coordinate system, $O_{i}(u_{0},v_{0})$ is the coordinate of the projected position of the camera’s optical axis in the image plane. $P(x_{w},y_{w},z_{w})$ is a certain point where the structured light plane intersects the surface of the object to be measured, and the intersection of the straight line $PO_{c}$ and the image plane is set to p(u,v). According to the perspective projection relationship of the pinhole imaging model, the mathematical model of the line-structured light 3D scanning system is established, as shown in Equation (1):(1)$$\lambda \left[\begin{array}{c} u\\ v\\ 1 \end{array}\right]=\left[\begin{array}{ccc} f_{x} & 0 & u_{0}\\ 0 & f_{y} & v_{0}\\ 0 & 0 & 1 \end{array}\right]\left[\begin{array}{cc} \begin{array}{l} R\\ 0^{T} \end{array} & \begin{array}{l} T\\ 1 \end{array} \end{array}\right]\left[\begin{array}{c} x_{w}\\ y_{w}\\ z_{w}\\ 1 \end{array}\right]$$
where λ is the scale factor; (u,v) denotes the coordinates in the pixel coordinate system; $f_{x}$ and $f_{y}$ denote the changing relationship between the image coordinate system and the pixel coordinate system; $\left(u_{0},v_{0}\right)$ denotes the coordinates of the projection of the optical axis of the camera lens in the pixel coordinate system; R is the 3×3 rotation matrix; T is the 3×1 translation matrix, and both represent the transformational relationship between the camera coordinate system and the world coordinate system.

#### 2.1.2. Point Cloud Pre-Processing

In the process of point cloud collection, due to the influence of external conditions and scanning equipment [27], the collected point clouds often contain many noise and outlier points which are not related to the surface information of the object and are usually useless. These will affect the measurement accuracy of the data, and the calculation process will take a long time [28]. The outliers are sparsely distributed in space, and statistical filtering [29] is effective for eliminating points with an obvious distribution. Therefore, according to the characteristics of outliers, it can be defined as noise when a point cloud dataset is less than a certain density. Statistical analysis is performed on the average distance from each point to its nearest *k* points, and the distances of all points should show a Gaussian distribution whose shape is determined by the mean μ and standard deviation σ. Let the coordinate of the n in the point cloud be $P_{n}(X_{n},Y_{n},Z_{n}),$ the distance $S_{i}$ from this point to any point $P_{m}(X_{m},Y_{m},Z_{m})$ is calculated in Equation (2):(2)$$S_{i}=\sqrt{{\left(X_{n}-X_{m}\right)}^{2}+{\left(Y_{n}-Y_{m}\right)}^{2}+{\left(Z_{n}-Z_{m}\right)}^{2}}$$

The average of the distance $S_{i}$ can be expressed as Equation (3):(3)$$\mu =\frac{1}{n}\sum_{i=1}^{n}S_{i}$$

The standard deviation of the distance $S_{i}$ can be expressed as Equation (4):(4)$$\sigma =\sqrt{\frac{1}{n}\sum_{i=1}^{n}{\left(S_{i}-\mu \right)}^{2}}$$

Set *std* as the standard deviation multiple. The *k* and *std* need to be input during the implementation of the algorithm. When the average distance of the *k* neighbors of a point is within the standard range (μ−σ·std,μ+σ·std), the point is retained; otherwise, it is defined as an outlier.

Statistical filtering is performed on the raw point cloud. The calculations show that the outliers in the raw point cloud can be removed well, and relatively complete parts characteristics are retained when the *k* is 20 and the *std* is 3. As shown in Figure 3a, the raw point clouds of the synchronous belt are collected, and the number of the point clouds is 22,025. Figure 3b is the point cloud to remove outliers by statistical filtering. The number of the point clouds is 21,891, and 123 outliers are removed.

#### 2.1.3. Point Cloud Voxel Down-Sampling

Due to the huge amount of raw point clouds obtained by the 3D scanning equipment, even after statistical filtering, the number of the point cloud is still large [30]. To improve the measurement efficiency, it is necessary to simplify the point cloud. Point cloud simplification should reduce the size of the data and eliminate redundancy, maintain the basic shape of the object, and highlight the key features of the object [31], such as protrusions, depressions, etc.

We compared four commonly used point cloud down-sampling methods [32], analyzed the advantages and disadvantages of different methods, and selected an appropriate point cloud down-sampling method according to the actual measurement requirements. Taking the Bunny of Stanford University as the comparison model, four different methods were used to down-sample the Bunny model, the point cloud down-sampling effect is shown in Figure 4.

Figure 4a shows the curvature of the point cloud used as the down-sampling basis; the larger the curvature, the more down-sampling points. Figure 4b is the result of random down-sampling, randomly sampling a certain number of points in the raw point cloud. Figure 4c shows the result of uniform down-sampling, the point clouds were filtered by constructing a sphere with a specified radius, and the point closest to the center of a sphere was used as the output point of the down-sampling. Figure 4d shows the result of voxel down-sampling, by creating a three-dimensional voxel grid from the raw point cloud. the center of gravity of all points within each voxel was used to approximate the other in the voxel. These four methods have simplified point clouds, but the point cloud position of the surface contour of the object has been changed. To further study the positional relationship of the object surface contour point cloud, the same position was intercepted for the simplified point cloud. The position of the intercepted slice was the red point cloud, as shown in Figure 4, and the contour point cloud projections are shown in Figure 5.

Among the three down-sampling methods of (a–c), some point cloud contours are missing, and the distribution is scattered. While the voxel down-sampling point cloud is relatively complete and evenly distributed, the contour features of the object are preserved well. Therefore, we use voxel down-sampling to simplify the point cloud. The process of the method is as follows:

(1) In the coordinate set of point cloud data, find the value $X_{max},Y_{max},Z_{max}$ and $X_{min},Y_{min},Z_{min}$ on the three axes of X, Y and Z, obtain the side length of the minimum bounding box of the point cloud $l_{x},l_{y},l_{z}$ by Equation (5). Set the side length of the voxel to *r*. After that, it can find the size of the voxel grid $d_{x},d_{y},d_{z}$ from Equation (6).
(5)$$\{\begin{array}{c} l_{x}=X_{max}-X_{min}\\ l_{y}=Y_{\;max}-Y_{\;min}\\ l_{z}=Z_{max}-Z_{min} \end{array}$$
(6)$$\{\begin{array}{c} d_{x}=\left\lfloor l_{x}/r\right\rfloor \\ d_{y}=\left\lfloor l_{y}/r\right\rfloor \\ d_{z}=\left\lfloor l_{z}/r\right\rfloor \end{array}$$

(2) Calculate the barycenter of all points in each voxel to approximately represent other points in the voxel. As shown in Figure 6, all points in the voxel can be represented by a barycentric point. Assume that the set of the local points in each voxel is $P\{x_{i},y_{i},z_{i}\}(i=1,2\ldots k),$ the centroid point $P_{g}$ can be found by Equation (7):(7)$$P_{g}=\frac{1}{k}\sum_{i=1}^{k}\left(x_{i},y_{i},z_{i}\right)$$

The side length of the voxel grid is set according to the size of the point clouds. The longer the side length of the voxel, the more points contained in each voxel. The resulting point clouds are sparser, and the details of the object are less. As shown in Figure 7, the density of the point cloud of the synchronous belt was compared in different voxel grid side lengths. Measuring different point cloud densities found that when *r* was equal to 0.1 mm, the point cloud down-sampling effect was the best, which could not only meet the measurement accuracy requirements but also significantly improved the operation speed.

#### 2.1.4. Point Cloud Plane Fitting

At present, the most common and simple method for fitting a plane is the least squares fitting, but the fitting accuracy is easily affected by noise. The random sample consensus (RANSAC) algorithm [33] can reduce the influence of noise by the iterative fitting and improve the fitting accuracy. In the RANSAC algorithm, first, a minimum number of random points are selected to define the geometric model of interest. The parameters of the model equation are calculated using the randomly selected points. The model parameters are applied to all points, whether they fit the model or not. If the appropriate number of points is found for the model to be defined, the parameters of the model and the error rate are calculated to check whether the established model is the best one. The algorithm runs iteratively until the best model is reached [34]. The process is as follows:

(1) In this work, the feature model of interest is a plane. It can be defined using at least three points which are randomly selected from the point cloud. Equation (8) is as follows:(8)a·x+b·y+c·z+d=0
where *a, b, c, d* are the parameters of the plane, and (*x, y, z*) are the 3D coordinates of any point.

(2) Calculate the distance $d_{i}$ from the remaining point to the plane by Equation (9):(9)$$d_{i}=\frac{\left|ax_{i}+by_{i}+cz_{i}+d\right|}{\sqrt{a^{2}+b^{2}+c^{2}}}$$
where $\left(x_{i},y_{i},z_{i}\right)$ denotes the remaining point of the point cloud.

If the distance $d_{i}$ is less than the threshold, this point is regarded as a point in the plane, and the number of interior points under the plane is counted and recorded.
(10)$${\left(d_{i}\right)}_{stay}\leq T$$
where *T* denotes the thresholds for segmenting the points of the extracted planes.

(3) Continue with steps 1~2. If the number of points in the plane exceeds the maximum number of interior points, recalculate the plane parameters with the saved interior points;

(4) Repeat steps 1 to 3 until there are no points that meet the conditions, and finally calculate the plane parameters again to obtain the best plane model.

According to the synchronous belt measurement plan, it is necessary to solve and fit plane A and plane B. The fitting result is shown in Figure 8. The red regions are the underside of the synchronous belt, and the blue regions are the point cloud of the tooth’s bottom surface. After the algorithm fitting, the plane A and plane B models are obtained.

#### 2.1.5. Point Cloud Curve Fitting

To improve the measurement accuracy of the synchronous belt tooth surface arc, this paper proposes a surface feature extraction method based on the point cloud normal vector. Based on the normal vector of the point cloud [35], set a reference vector and calculate the angle value between the normal vector of each point in the point cloud set and the reference vector; determine its value and complete the adjustment of the tooth surface arc of the synchronous belt. The process of feature extraction is as follows:

(1) Normal vector estimation. Firstly, estimate the normal vector of the point cloud, any point $p_{i}$ in the point cloud, search for its nearest neighbors k, and then use the least squares method to calculate the local plane $P_{l}$ of this point. Equation (11) is as follows:(11)$$P_{l}(\vec{n},d)=\underset{\left(\vec{n},d\right)}{\arg\min}\sum_{i=1}^{k}{\left(\vec{n}\cdot p_{i}-d\right)}^{2}$$
where $\vec{n}$ is the normal vector of the plane $P_{l},$ and *d* is the distance from $P_{l}$ to the origin of the coordinate.

Then, it can be considered that the normal vector of the plane fitted by *k* nearest points is approximated as the normal vector of the point $p_{i}$, and the estimated normal vector of the plane $P_{l}$ is obtained by performing PCA (principal component analysis). It can decompose the eigenvalues of a covariance matrix M by Equation (12). The eigenvector corresponding to the smallest eigenvalue of the matrix is the normal vector of the plane $P_{l},$ which is also the normal vector of the point $p_{i}.$
(12)$$M=\frac{1}{k}\sum_{i=1}^{k}\left(p_{i}-p_{0}\right){\left(p_{i}-p_{0}\right)}^{T}$$
where $p_{0}$ is the centroid of the *k* points.

(2) Normal vector orientation. The normal vector is ambiguous [36]. The previous calculation only obtains the straight line where the normal vector is located but does not determine the final direction of the normal vector [37], as shown in Figure 9a. Before using the normal vector angle value for surface feature extraction, it is necessary to ensure normal vector consistency. Set a reference vector $\vec{a},$ and then traverse the normal vectors $\overset{\to }{b_{i}}$ of other points. If $\vec{a}\cdot \overset{\to }{b_{i}}<0,$ then the vector direction is opposite. Flip the normal vector $\overset{\to }{b_{i}},$; otherwise, it remains unchanged. The adjustment result is shown in Figure 9b.

(3) The region of interest curve. The tooth surface profile of the synchronous belt is composed of two arcs, which cannot be directly fitted and solved. Therefore, it is necessary to extract the effective arc area of the tooth surface profile to improve the measurement accuracy. In the feature area, the deviation of the normal vector direction of the point cloud is large, and the angle of the sampling point can be used for identification and extraction. The extraction method is as follows: based on the consistency of the normal vector of the point cloud, Equation (13) can calculate the angle θ between the reference vector $\vec{a}$ and the normal vector $\overset{\to }{b_{i}}$ of the sampling point and set the discrimination threshold. If θ>f, mark the point as a curve feature point dataset; otherwise, mark it as a non-curve feature point set.
(13)$$\theta =\cos\langle \vec{a},\vec{b}\rangle =\frac{\vec{a}\cdot \vec{b}}{\left|\vec{a}\right|\cdot \left|\vec{b}\right|}$$

After experimental verification, when the threshold is set to f=0.65, the error of the measurement result is the smallest, and the segmentation effect is shown in Figure 10. The red area is the region of the interest curve.

(4) The tooth surface arc fitting. Through the above steps, the effective arc area of the tooth surface profile is extracted, but the point cloud dataset is spatial three-dimensional data. The fitting accuracy is not high, and the efficiency is low. To fit the contour of the surface accurately and efficiently, the slicing method [38] is introduced to process the point cloud datasets and realize the dimensionality reduction in the point cloud datasets. The process is as follows: the essence of point cloud slicing is the intersection of a set of planes and point cloud datasets. By setting the slice width *d*, the point clouds are divided into a certain number of point cloud datasets and then projected to the plane to obtain a two-dimensional point map of the object outline, as shown in Figure 11a. The least squares method [39] is used to improve the fitting accuracy of slices, and the fitting process is as follows:

(1) Build the circle Equation (14):(14)$${\left(x-A\right)}^{2}+{\left(y-B\right)}^{2}=R^{2}$$

(2) The difference between the distance from the point $\left(X_{i},Y_{i})i\in (1,2\cdots N\right)$ to the center of the circle and the radius is shown in Equation (15):(15)$$\begin{array}{ll} d_{i} & ={\left(X_{i}-A\right)}^{2}+{\left(Y_{i}-B\right)}^{2}-R^{2}\\ & ={X_{i}}^{2}+{Y_{i}}^{2}-2AX_{i}-2BY_{i}+A^{2}+B^{2}-R^{2} \end{array}$$

(3) Let a=−2A, b=−2B, $c=A^{2}+B^{2}-R^{2},$ substitute them into Equation (14) and simplify, then find Equation (16).
(16)$$d_{i}={X_{i}}^{2}+{Y_{i}}^{2}+aX_{i}+bY_{i}+c$$

(4) Let $Q(a,b,c)={\sum d_{i}}^{2},$ because Q(a,b,c) is much greater than 0. The function has a minimum value greater than or equal to 0 and a maximum value of infinity. Find the partial derivative for *a, b,* and *c*, and set the partial derivative equal to 0. The following can be obtained:(17)$$\{\begin{array}{c} \frac{\partial Q(a,b,c)}{\partial a}=\sum 2({X_{i}}^{2}+{Y_{i}}^{2}+aX_{i}+bY_{i}+c)X_{i}=0\\ \frac{\partial Q(a,b,c)}{\partial b}=\sum 2({X_{i}}^{2}+{Y_{i}}^{2}+aX_{i}+bY_{i}+c)Y_{i}=0\\ \frac{\partial Q(a,b,c)}{\partial c}=\sum 2({X_{i}}^{2}+{Y_{i}}^{2}+aX_{i}+bY_{i}+c)=0 \end{array}$$
solve Equation (17) to obtain the extreme point, compare the function values of all extreme points. The parameter corresponding to the minimum value is the best fitting circle parameter, and the fitting diagram is shown in Figure 11b. The fitting circle is close to the sampling point, which indicates the fitting result is well.

## 3. Results

### 3.1. Test Environment

The composition of the synchronous belt tooth profile experimental is shown in Figure 12. The collection of point clouds is completed by the line laser profile sensor. According to the belt thickness range of the synchronous belt, the distance between the sensor and the mobile platform is 174 mm; the moving speed of the mobile platform is 350 mm/s; The line laser profile sensor is connected to the computer through Ethernet, and the point cloud datasets are transferred to the computer for the following process. The software platform and library are Python V3.8 and Open3D point cloud library V0.15.0.

### 3.2. Measurement Data

The arc tooth synchronous belt is placed on the platform, and the point cloud on the surface of the arc tooth synchronous belt is obtained by the line structured light sensor. The raw point clouds are preprocessed, and the plane and region of the interest curve are obtained by dividing the point cloud. As in the measurement plan in Section 2, the tooth profile parameter measurement of the synchronous belt is completed.

To verify the accuracy and stability of our method, the result of the projection and the manual method were compared. Taking the manual measurement data as the standard, the error analysis of the projection method and the method in this paper is carried out. The measurement results and errors are shown in Table 1. It can be seen from the table that the error of our method is smaller than that of the projection method, the measurement data are closer to the manual measurement data, and the error range meets the measurement requirements.

## 4. Discussion

For the measurement data of synchronous belts, manual measurement is closer to the actual size, but its measurement speed is slow, the stability is poor, and it cannot work for a long time, which is difficult to process and produce automatically. It is gradually replaced by other methods.

The projection method places the light source on the back of the object. It is suitable for the detection of the outline of the opaque object. However, it is more sensitive to the intensity of the light source. When the light source is weak, the edge contour of the object is affected by burrs, and the contour of the object cannot be accurately extracted; when the light source is strong, some details of the contour are corroded, and the contour features are lost, and the actual size of the object cannot be measured accurately. As shown in Figure 13 and Figure 14, the Pb and ht measured by the projection method are slightly larger than others. In contrast, it can be seen from Figure 15 that the measurement data of the hs are slightly smaller. This is due to the strong light source that depicts the projection profile as smaller than the actual contours.

With our method, the surface profile information of the object is obtained by the line-structured light sensor, and the tooth profile parameters of the synchronous belt are measured based on the point cloud segmentation and fitting technology. Since the influence of the light source does not need to be considered in the process of point cloud acquisition, the measurement data are closer to the actual size, with less error than the projection method, and the stability is better than that of the manual method, which can meet the requirements of continuous and stable work and is conducive to the realization of automatic processing and production work.

## 5. Conclusions

In this work, a 3D point cloud measurement system for synchronous belts is built using a line-structured light sensor. Firstly, the contour point clouds of the synchronous belt are collected. To improve the accuracy and efficiency of measurement, the raw point clouds are filtered, and the number of the point cloud is reduced. Then, referring to the measurement plan of the synchronous belt, the point clouds are divided into plane and arc areas. In the plane areas, it can effectively reduce the fitting error by using the RANSAC algorithm. To obtain the effective area of the arc, a segmentation method based on a normal vector is proposed, which can effectively segment the target area. Subsequently, a curve fitting model suitable for the arc tooth profile is established by the least squares method. Finally, the parameters of each tooth are calculated. The experimental results show that the method in this paper can accurately and quickly detect the tooth profile parameters of the synchronous belt. The average errors of this method in parameters are less than 0.03 mm. This method provides a useful solution with a much smaller degree of error in the stability of the tooth profile. It is an obvious advantage over the classic, common methods. Therefore, the detection scheme has great application prospects and promotion value. The main work in this paper focuses on point cloud noise reduction, point cloud simplification, point cloud segmentation, and size measurement. However, the inspection of surface defects on the synchronous belt has not been thoroughly studied. Therefore, further research can be carried out on surface defects.

## Figures and Tables

**Figure 1 sensors-22-06372-f001:**
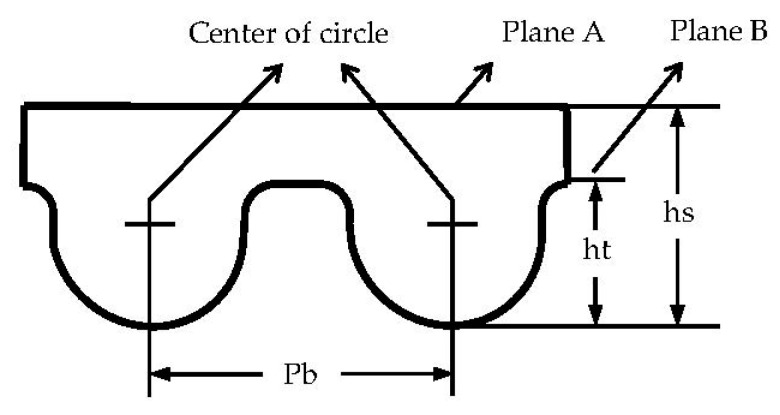
Synchronous belt parameters.

**Figure 2 sensors-22-06372-f002:**
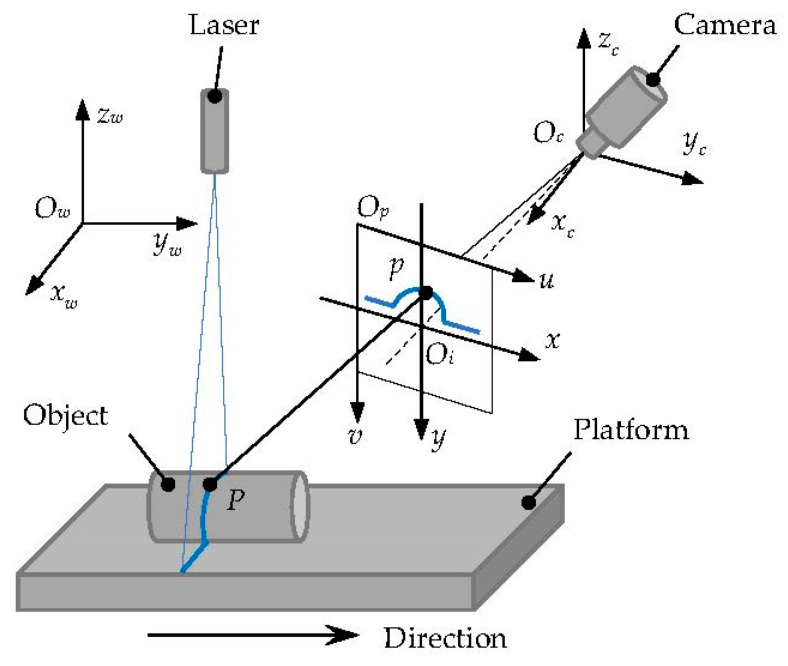
Point cloud acquisition system model.

**Figure 3 sensors-22-06372-f003:**
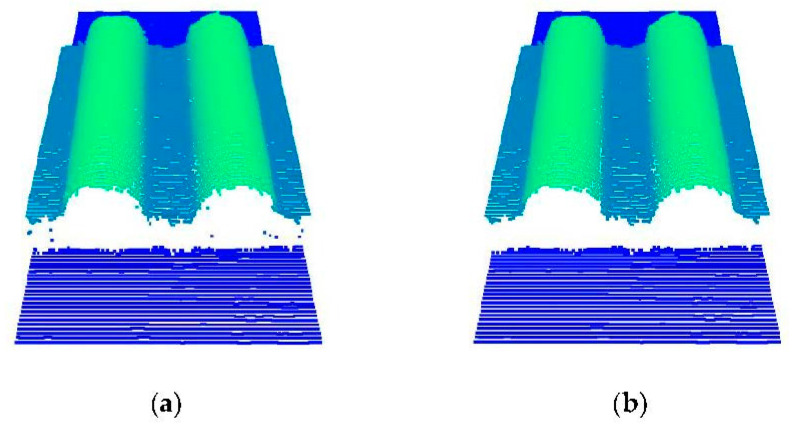
Point cloud before and after statistical filtering: (**a**) raw point cloud and (**b**) point cloud after statistical filtering.

**Figure 4 sensors-22-06372-f004:**
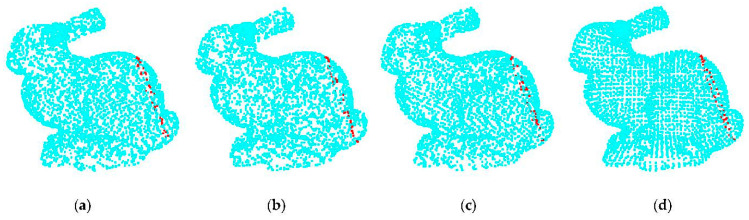
Point cloud down-sampling effect: (**a**) curvature down-sampling; (**b**) random down-sample; (**c**) uniform down-sample; and (**d**) voxel down-sample.

**Figure 5 sensors-22-06372-f005:**
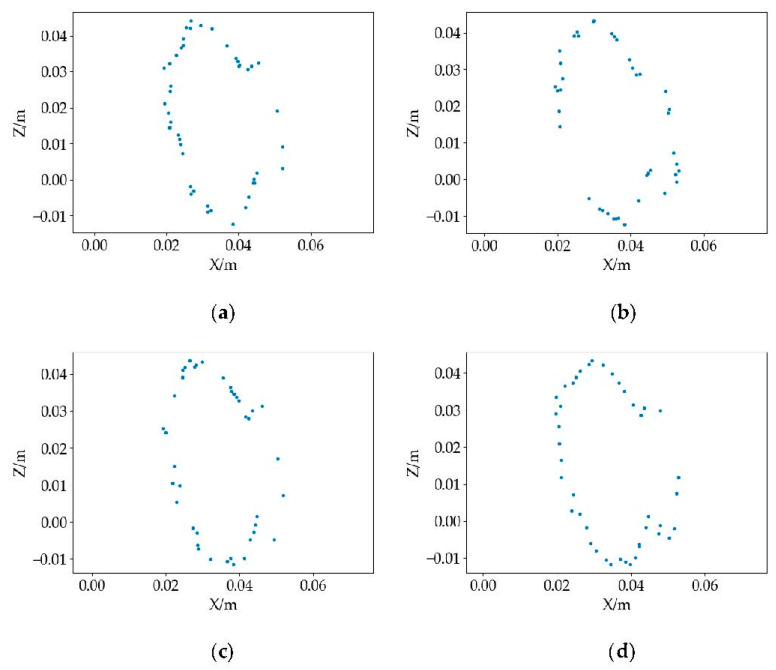
Contour point cloud projection: (**a**) curvature down-sampling; (**b**) random down-sample; (**c**) uniform down-sample; and (**d**) voxel down-sample.

**Figure 6 sensors-22-06372-f006:**
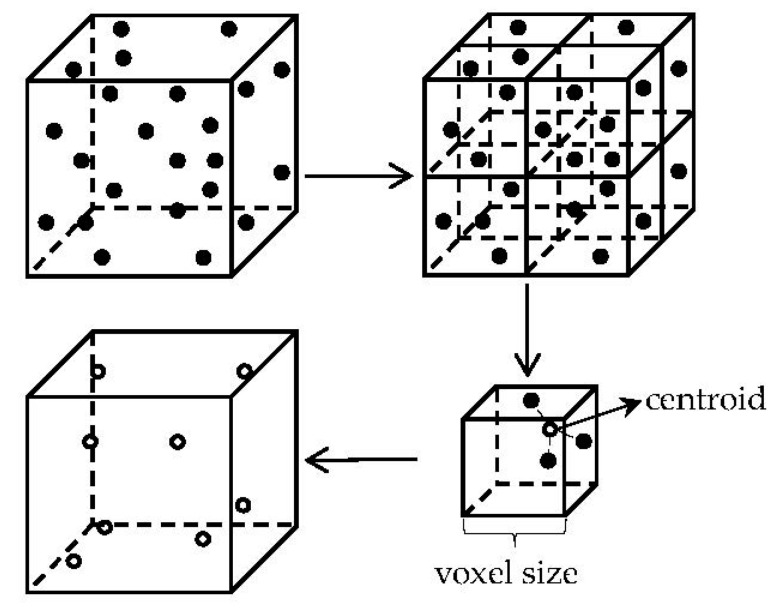
Process of voxel down-sampling.

**Figure 7 sensors-22-06372-f007:**
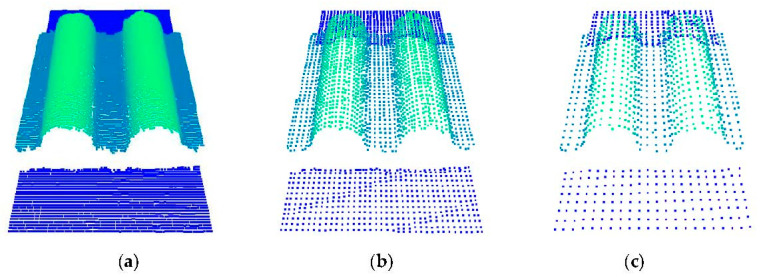
Voxel down-sampling of point cloud with different side lengths: (**a**) *r* = 0.1 mm; (**b**) *r* = 0.3 mm; and (**c**) *r* = 0.5 mm.

**Figure 8 sensors-22-06372-f008:**
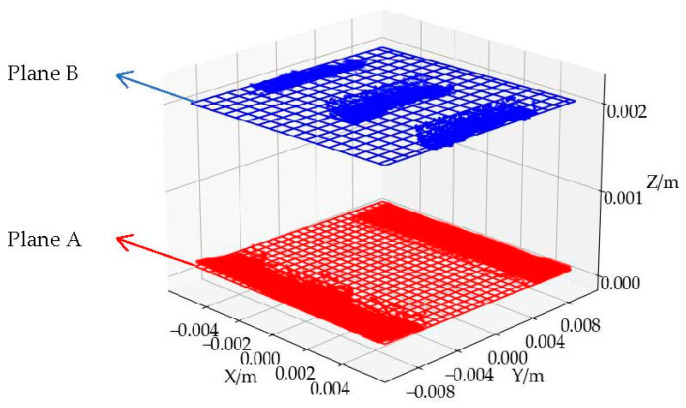
Fitting result of plane A and plane B.

**Figure 9 sensors-22-06372-f009:**
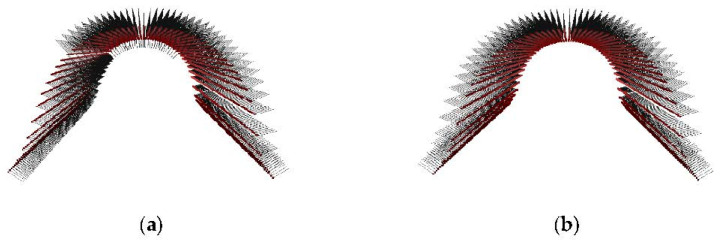
Normal vector orientation: (**a**) ambiguity of point cloud normal vector and (**b**) consistency of point cloud normal vector.

**Figure 10 sensors-22-06372-f010:**
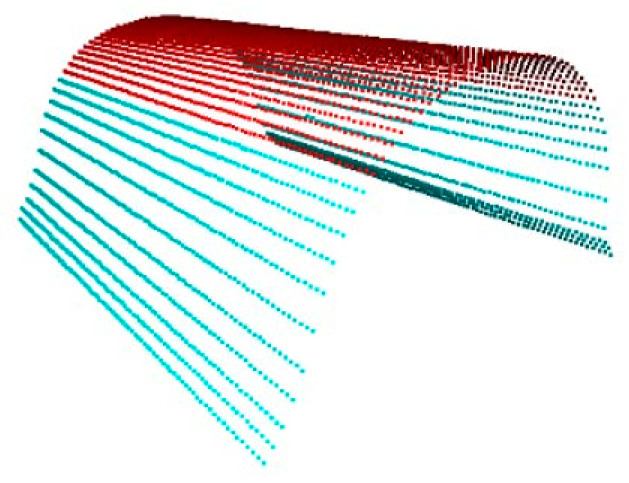
Region of Interest Curve.

**Figure 11 sensors-22-06372-f011:**
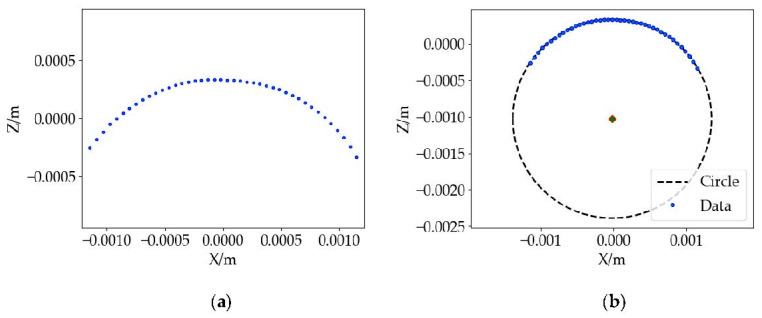
Tooth surface arc fitting: (**a**) contour point cloud projection, and (**b**) schematic diagram of least squares fitting circle.

**Figure 12 sensors-22-06372-f012:**
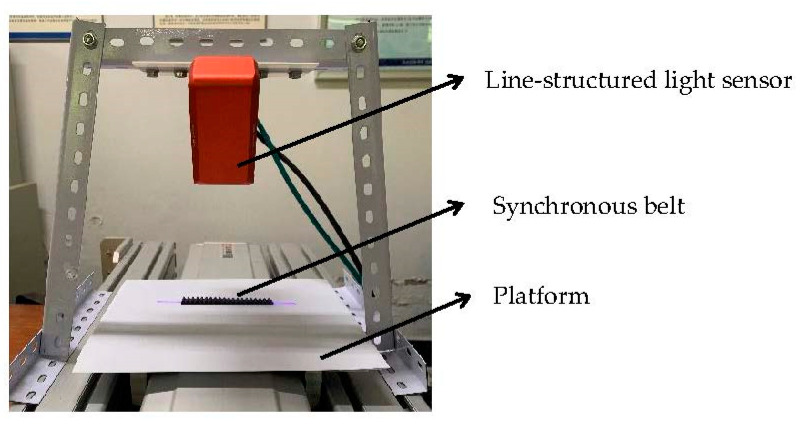
Synchronous belt tooth profile measurement system composition.

**Figure 13 sensors-22-06372-f013:**
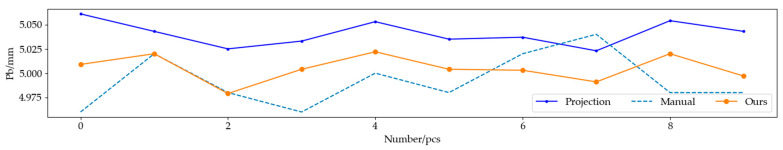
Graph of Pb measurement data.

**Figure 14 sensors-22-06372-f014:**
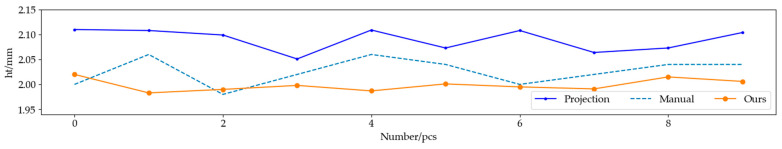
Graph of ht measurement data.

**Figure 15 sensors-22-06372-f015:**
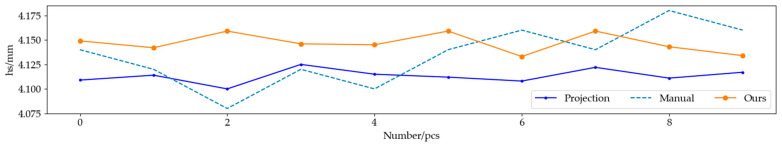
Graph of hs measurement data.

**Table 1 sensors-22-06372-t001:** Measurement data of tooth profile parameters.

Number	Projection Method			Manual Method			Our Method		
	Pb/mm	ht/mm	hs/mm	Pb/mm	ht/mm	hs/mm	Pb/mm	ht/mm	hs/mm
1	4.960	2.000	4.140	5.061	2.110	4.109	5.009	2.020	4.149
2	5.020	2.060	4.120	5.043	2.108	4.114	5.020	1.983	4.142
3	4.980	1.980	4.080	5.025	2.099	4.100	4.979	1.990	4.159
4	4.960	2.020	4.120	5.033	2.051	4.125	5.004	1.998	4.146
5	5.000	2.060	4.100	5.053	2.109	4.115	5.022	1.987	4.145
6	4.980	2.040	4.140	5.035	2.073	4.112	5.004	2.001	4.159
7	5.020	2.000	4.160	5.037	2.108	4.108	5.003	1.995	4.133
8	5.040	2.020	4.140	5.023	2.064	4.122	4.991	1.991	4.159
9	4.980	2.040	4.180	5.054	2.073	4.111	5.020	2.015	4.143
10	4.980	2.040	4.160	5.043	2.104	4.117	4.997	2.006	4.134
Average	4.992	2.026	4.134	5.041	2.090	4.113	5.005	1.999	4.147
Error				0.049	0.064	0.021	0.013	0.027	0.013

## Data Availability

The data used to support the findings of this study are included within the article.

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
