# Peer review of "Research on Measurement of Tooth Profile Parameters of Synchronous Belt Based on Point Cloud Data"

_sensors, 2022, doi:10.3390/s22176372_

Round 1

Reviewer 1 Report

The authors proposed an interesting procedure to solve the novel problem in the field of dental health.

The procedure proposed by the authors is based on an original, pragmatic algorithm and is subsequently validated on a significant set of measurements. The method provides a useful solution with a much smaller degree of error in the stability of the tooth profile. It is an obvious advantage over the classic, common methods. The statistical apparatus is used correctly, and then they are filtered through relevant tests.

Reviewer 2 Report

Numbers are lines in the article:

97. Before The Oc put dot not comma.

107. Put where at the beginning of the line not after tab.

120. Fig.2  needs corrections. Quality is not so good. Lines are crooked, arrows are missing.

137, 139. Why capital letter (Equation) not equation? The note applies to the all article.

139. After std put "comma" not "space" and "comma"

164. After Z put "dot" not "space" and "dot"

173. After Figure1 and others. Put "space" between number of the fig. and beginning of the sentence.

209. Figure is not legible to me.

253. Move the description of the drawing to the page where drawing is.

Reviewer 3 Report

The authors proposed a method for measuring the tooth profile parameters of the synchronous belt based on point cloud data to realize the automatic detection of the tooth profile parameters of the synchronous belt, improve the detection efficiency, and reduce the influence of human factors. In this study, the line structured light sensor is used to obtain the point cloud data of the synchronous belt.  The results look encouraging and motivating. But there are still some contents, which need be revised in order to meet the requirements of publish. A number of concerns listed as follows:

(1)   The theoretical background of the proposed method is adequately detailed in the paper.

(2)   To explore Comparative results with existing approaches/methods relating to the proposed work. The method/approach in the context of the proposed work should be written in detail.

(3)   In the introduction section, you should give the novelty and the contributions of your works.

(4)   The literature review is poor in this paper. You must review all significant similar works that have been done.  For example, 10.1016/j.isatra.2021.07.017 ;10.1109/JSTARS.2021.3059451 ; 10.3390/agriculture12060793 10.1007/s10489-022-03719-6 and so on.

(5)   A deep and accurate comparison of your approach with the current state of the art should be added in your manuscript.

(6)   The authors need to interpret the meanings of the variables

(7)   Lines 265-269, how to obtain these expressions? According to……

(8)   Simplify the point cloud? Why is the voxel down-sampling method used in this paper?

(9)   At Line 179, the number of point clouds is 14359. The authors should provide how to set?

(10) There are some grammatical mistakes and typo errors. please proof read from native speaker.

Round 2

Reviewer 3 Report

According to the revised paper, I have appreciated the deep revision of the contents and the present form of this manuscript.  There is little content, which need be revised according to the comment of reviewer in order to meet the requirements of publish. A number of concerns listed as follows:

(1) The authors need to interpret the meanings of the variables.

(2) Please highlight your contributions in introduction.

(3) How to determine these parameters? The author should give a detailed explanation.

(4) Conclusion: What are the advantages and disadvantages of this study compared to the existing studies in this area?

(5) The inspiration of your work must further be highlighted. Some suggested recent literatures in the previous comment should add in the revised paper.

(6) Further correct typological mistakes and mathematical errors.

(7) More equations are necessary to explain the proposed method.

I hope that the authors can carefully and further revise this manuscript according to the reviewer comments in order to meet the requirements of publish.
